# Unravelling the Complexity of Irregular Shiftwork, Fatigue and Sleep Health for Commercial Drivers and the Associated Implications for Roadway Safety

**DOI:** 10.3390/ijerph192214780

**Published:** 2022-11-10

**Authors:** Jessica Erin Mabry, Matthew Camden, Andrew Miller, Abhijit Sarkar, Aditi Manke, Christiana Ridgeway, Hardianto Iridiastadi, Tarah Crowder, Mouyid Islam, Susan Soccolich, Richard J. Hanowski

**Affiliations:** 1Division of Freight, Transit, and Heavy Vehicle Safety, Virginia Tech Transportation Institute, Blacksburg, VA 24061, USA; 2Faculty of Industrial Technology, Institut Teknologi Bandung, Bandung 40132, Indonesia

**Keywords:** fatigue, sleep, safety

## Abstract

Fatigue can be a significant problem for commercial motor vehicle (CMV) drivers. The lifestyle of a long-haul CMV driver may include long and irregular work hours, inconsistent sleep schedules, poor eating and exercise habits, and mental and physical stress, all contributors to fatigue. Shiftwork is associated with lacking, restricted, and poor-quality sleep and variations in circadian rhythms, all shown to negatively affect driving performance through impaired in judgment and coordination, longer reaction times, and cognitive impairment. Overweight and obesity may be as high as 90% in CMV drivers, and are associated with prevalent comorbidities, including obstructive sleep apnea, hypertension, and cardiovascular and metabolic disorders. As cognitive and motor processing declines with fatigue, driver performance decreases, and the risk of errors, near crashes, and crashes increases. Tools and assessments to determine and quantify the nature, severity, and impact of fatigue and sleep disorders across a variety of environments and populations have been developed and should be critically examined before being employed with CMV drivers. Strategies to mitigate fatigue in CMV operations include addressing the numerous personal, health, and work factors contributing to fatigue and sleepiness. Further research is needed across these areas to better understand implications for roadway safety.

## 1. Introduction

For the general working population, fatigue is characterized by a state of feeling tired and/or sleepy as a result of prolonged mental or physical work, overexertion, extended periods of anxiety, exposure to harsh environments, boredom, and/or sleep loss [1,2]. Distinguishing between task- and sleep-related fatigue gives context to the etiology of the condition; however, the presentation of symptoms are similar and problematic for safety-sensitive work populations that require vigilance and cognitive acuity [3]. Within the transport sector, a more precise operational definition is needed, describing the etiology of fatigue and sleepiness, effects on driving performance and crashes, and strategies to ameliorate these problems (e.g., [4,5,6,7]). Multiple factors influence worker fatigue, including time awake, time of day, workload, and sleep quality and quantity [1,8]. Fatigue is a significant problem for commercial motor vehicle (CMV) drivers, largely due to high-demand jobs, long duty periods, disruptions of circadian rhythms, and accumulative sleep debt [1,8]. Fatigue can impact CMV drivers’ health and safety, as well as the safety of those they work with and other road users [1,9,10]. This narrative review highlights the literature and key findings regarding shiftwork, fatigue, and CMV driver sleep health and the implications for roadway safety. The literature was sourced using PubMed, Embase, PsycInfo, Scopus, and Google Scholar databases, with a focus on articles published from 2000 and later (though exceptions were made for older publications with scientific relevance). Keywords and search terms are included in Table 1. Specific topics discussed include: (i.) impacts of lifestyle and behavioral factors on CMV driver health; (ii.) impacts of shiftwork and irregular scheduling on sleep; (iii.); health and medical conditions that can impact fatigue; (iv.) impacts of fatigue on driving and crash risk; (v.) screening and diagnostic tools to assess fatigue; and (vi.) solutions, challenges and research needs, including tools, technologies, and safety programs to mitigate fatigue in CMV operations.

## 2. Impacts of Lifestyle and Behavioral Factors on Commercial Driver Health

CMV drivers may experience adverse health outcomes due to the nature of the work environment, extended time spent on the road, and perceived barriers to a healthy lifestyle [11]. Previous research has linked multiple lifestyle and work-related factors typically experienced by CMV drivers to chronic comorbidities [11,12,13,14,15,16,17,18,19,20]. Drivers function daily in environments that present challenges for healthy behaviors—including truck stops, fleet terminals, shipping and receiving warehouses, and truck cabins—where it is often difficult to find facilities and space for recommended exercise and healthy eating options [8,11,12,21,22]. Thus, CMV drivers tend to lead sedentary lifestyles [23] and may be complacent or inattentive toward healthy behaviors. Long working hours, irregular sleep schedules, and lifestyle factors increase the chances of CMV drivers being susceptible to health risks such as obesity, cardiovascular disease, and metabolic disorders [12,18,24,25,26,27,28,29]. In addition, drivers are subjected to continuous vibrations in their cab while driving, increasing risks for lower back pain and musculoskeletal disorders [11]; those who obtain the bulk of their sleep at truck stops are also exposed to pollutants and irritants [30]. The consequences of lifestyle and work stressors may cause CMV drivers to have a reduced life expectancy compared to the general U.S. male population [31,32]. Furthermore, extended periods of time away from home may limit drivers’ access to health care and use of health insurance [8,33].

CMV drivers function within strict government regulations that control hours of service (HOS) for drive and on-duty time [34]. HOS regulations restrict the number of hours that commercial drivers are allowed to work in a 24-hour day or a 7-day week, ensuring they acquire sufficient rest to reduce fatigue-related crashes [19]. They are expected to perform under stringent scheduling while usually being paid by the mile; time pressures may contribute to drivers relying on fast food and quick, on-the-go meals and snacks, limiting opportunities to make healthy eating choices [8,21,35]. The consistent consumption of high-calorie, low-nutrient foods can lead to overweight and obesity. Excess weight affects 90% of CMV drivers and has numerous associated comorbid conditions, including cardiovascular disease and sleep, metabolic, and psychological disorders [8,13,17,23,26,33,35,36,37,38]. Additionally, evidence suggests that fatigue levels positively correlate with work stress, life stress, depression, and anxiety while having a negative relationship with life satisfaction, work happiness, and work satisfaction [3,39,40]. In summary, many lifestyle and behavioral factors inherent in the CMV industry contribute to the high prevalence of obesity and related comorbidities observed in this population.

## 3. Impacts of Shiftwork and Irregular Scheduling on Sleep Physiology and Circadian Rhythms

Multiple factors influence worker fatigue, including time awake, time of day, workload, driving hours, working hours, and sleep quality and quantity [1,8,29,40,41]. Lacking sleep, sleep restriction, poor-quality sleep, and variations in circadian rhythms have been shown to negatively affect driving performance through impaired judgment and coordination, longer reaction times, and cognitive impairment that impacts memory and the ability to retain information [42,43,44,45]. For example, one study found that heavy vehicle drivers who averaged less than 5 h of sleep experienced a higher frequency of fatigue-related safety critical events compared to drivers that obtained 7–8 h of sleep per day [46]. Circadian rhythms contribute to fatigue-related crashes, with a higher proportion of sleep-related crashes occurring during the early morning hours (2–6 a.m.), and during early afternoons (2–4 p.m.), the latter being known as the “post-lunch dip” [10,47]. Fatigue level and impaired driving performance are impacted by the combined contributions of driving duration and circadian rhythms [48].

Shift work is also associated with impaired alertness and performance due to sleep loss and circadian misalignment [49]. Shift-work schedules often involve extended episodes of wakefulness, early start and/or late end times, which may impact sleep duration and increase sleep–wake disturbances [50]. The combined effect of these circadian and sleep-related factors can impair alertness and job performance [51,52]. Individual tolerance to shift work and irregular work hours is complex, as it is affected by work hours, shifts, rest periods, predictability of work schedules, and individual differences. Split shifts (assigned shifts changing daily or weekly) can lead to even greater risk for developing fatigue when work and sleep schedules fluctuate, as shown in self-report data from heavy vehicle operators following unexpected split shifts [53,54,55]. Working on atypical or irregular shifts, as CMV drivers often do, has important implications for not only sleep health but also crash risk, impairment, and public safety [43].

## 4. Medical Conditions That Can Impact Fatigue and Sleepiness

As noted previously, overweight and obesity—which may be as high as 90% in CMV drivers—and associated comorbidities, including obstructive sleep apnea (OSA), hypertension, cardiovascular and metabolic disorders, could have significant implications for fatigue and safety among this population [17,19,23,26,33,36,37,56,57,58,59]. In a naturalistic driving study with CMV drivers, Wiegand et al. [60] found obese CMV drivers were 1.2 to 1.7 times more likely to be fatigued while driving and were at 1.37 times greater risk for being involved in a safety-critical event than non-obese drivers; obese drivers were almost twice as likely to be fatigued while involved in an at-fault safety-critical event [60,61].

OSA is prevalent, but also severely undiagnosed among CMV drivers. Estimates suggest 25–40% of CMV drivers have undiagnosed OSA [62,63,64]. The prevalence of OSA increases with age and BMI, both of which are increasingly common in the CMV driver workforce, as drivers are retiring later and/or entering the industry later in life. A lack of routine screening and limited diagnostic sleep study facilities contribute to high rates of undiagnosed OSA; furthermore, common signs and symptoms of OSA may be difficult to self-diagnose or may be attributed to other issues. CMV drivers may be hesitant to disclose symptoms or undergo voluntary testing, knowing that a diagnosis will force them into positive airway pressure, or PAP, treatment or end their career [37].

Vgontzas et al. argues that obesity is associated with objective and subjective sleepiness in the absence of sleep OSA or sleep disorders, and that metabolic disturbances associated with obesity, including diabetes and insulin resistance, are primary determinants of subjective sleepiness [65]. Fatigue and sleepiness can exacerbate obesity and associated comorbidities through mediating factors, including reductions in physical activity, increases in appetite, hormone dysregulation, and sympathetic overdrive [29,66,67,68,69,70]. Studies indicate that OSA is associated with decreased physical activity and increased preference for calorie-dense foods that are high in fat and carbohydrates, as well as hormone dysregulations that can stimulate appetite to exacerbate obesity through alterations in energy balance and metabolism [71]. Hypertension, impacting up to 40% of CMV drivers [23,24], is linked to OSA through mechanisms of action in the sympathetic nervous system that elevate heart rate and blood pressure in response to repetitive and stressful sympathetic activations. Hickman et al. determined that hypertension was the second most predictive factor (after BMI) for having OSA among a cohort of 21,000 CMV drivers [23]. In summary, there is a significant relationship between comorbidities associated with obesity in this population and driver fatigue and work safety.

## 5. Impacts of Fatigue and Sleepiness on Driving Performance and Crashes

Although the physiological causes and factors can differ for fatigue and sleepiness, their outcome on driver safety is similar. As cognitive and motor processing declines, driver performance decreases, and the risk for errors, near crashes, and crashes increases [72,73]. Fatigued driving is a serious traffic safety hazard due to the complex interaction of unavoidable biological phenomena (circadian rhythm) and health issues, driving environment, and monotonous stretches of highways [74]. Fatigue-related crashes resulting in injury and death are estimated at roughly USD 109 billion annually (without property damage cost) [75]. The Large Truck Crash Causation Study (LTCCS) found that CMV driver fatigue was associated with 13% of crashes [76]. This estimate is thought to be low, as it is difficult to determine post hoc whether fatigue contributed to or was a primary contributing factor in a crash [17]. In 2015, 400,000 truck crashes, resulting in 87,000 injuries and 4000 fatalities, were associated with drivers being “asleep or fatigued.”

Howard et al. observed a relationship between crash risk and subjective measures of chronic sleepiness, discovering the sleepiest 5% of drivers were twice as likely to have been involved in a recent, self-reported crash [77]. In a case–control study with 200 long-haul CMV drivers, Meuleners et al. noted fatigue-related factors were associated with crash risk, with risk being almost 5 times higher when driving between midnight and 5:59 a.m., and increased risk when more than 2 h had passed since the last break during the trip [64]. Chu examined the contributing risk factors for severe crashes involving high-deck buses used for long-distance highway driving, finding that fatigued driving was among the driver variables that showed a greater percentage of leading to fatal and injury crashes compared to no-injury crashes [78]. Long-distance driving is more likely to cause driver fatigue, reduced alertness and concentration, and impaired driver skills, thus increasing the risk of serious injury. A cross-sectional study with bus drivers revealed that sleep problems were frequent among drivers and had a significant relationship with prior crash history [79]. Akkoyunlu et al. discovered a positive correlation between bus drivers’ subjective sleepiness and crash rate [80]. Pack et al. evaluated the role of prolonged short sleep and OSA in subjective and objective assessments of sleepiness and driving performance tasks in a sample of CMV drivers at low and high risk for OSA [81]. Subjective sleepiness was associated with shorter sleep durations, but not with increases in OSA severity. Similarly, objective sleepiness, performance lapses, and decreases in lane tracking were associated with shorter sleep duration. The authors suggested that insufficient sleep, independent of OSA, must be considered when addressing impairment. Prevalence of sleepiness and fatigue is high among CMV drivers and has important implications for safety among this population.

## 6. Screening and Diagnostic Tools to Assess Fatigue and Sleepiness

Assessments to determine and quantify the nature, severity, and impact of fatigue across a variety of environments and populations have been developed. While a myriad of subjective assessment techniques are available, commonly used measures within the driving context include the Karolinska Sleepiness Scale [82], Stanford Sleepiness Scale [83], Epworth Sleepiness Scale [84], Johns Drowsiness Scale [85] and Visual Analogue Scale [86]. Instruments such as the Pittsburgh Sleep Quality Index (PSQI) have also been developed to assess sleep quality, which is known to impact driver fatigue [87,88]. It should be noted, however, that these subjective measures can be challenging to administer, are sensitive to several potential biases [89], and may not correlate well with (more objective) neurophysiological indicators. Validated tools for sleep disorder screening, including the Berlin Questionnaire and the STOP-Bang, include both subjective and objective factors in their risk assessment algorithm [90,91]. As part of the risk management system, fatigue can be modeled and thus predicted based on fatigue-causing variables such as sleep–wake profile, circadian rhythms, and work schedule [92]. More objective measures of fatigue, such as the psychomotor vigilance task [93], have been used widely in road transport, as has the sustained attention test for the rail sector [50].

Physiological signals hold vital information about a person’s health and their affective nature. Heart rate, pulse rate, breathing rate, galvanic skin response, and blood pressure have been historically used to study human psychophysiology. This extends to driver health monitoring, which is an active field of research. Knowledge of a driver’s physical and cognitive state has potential for understanding key safety implications in roadway safety. Most of these physiological signals reflect the behavior of the autonomic nervous system (ANS), which controls involuntary behaviors, including heart rate, respiration rate, and blood pressure, which reflect an individuals’ psychophysiological condition, including stress, mood, and level of fatigue. ANS demonstrates the balance between sympathetic and parasympathetic responses. Therefore, measuring signals such as heart rate, blood pressure, and skin conductance can provide information on the state of the sympathetic and parasympathetic nervous systems. For example, skin galvanic response shows the electro-dermal activity that directly correlates to stress. Similarly, heart rate variability has been used to understand cognitive load [94]. Historically, a major challenge for real-time implementation of such measurement systems is that most of the devices are too cumbersome and intrusive for driving research. Recent years have seen enormous progress in wearable electronic devices [95,96] and remote measurement methods [97] that can measure physiological signals without interfering with driving activities. While many screening and diagnostic tools are available to assess fatigue, each method has benefits and limitations which must be considered relative to the population and specific research questions.

## 7. Solutions, Challenges, and Research Needs

Strategies to mitigate or reduce fatigue in CMV operations include addressing the numerous personal, health, and work factors contributing to fatigue and sleepiness. Further research is needed across these areas to better understand implications for roadway safety. Research may also support implementing fatigue detection technologies in fleet and industry operations that can predict, detect, and alert to operator fatigue [98]. Chapters 8–10 discuss research needs, challenges, and solutions.

## 8. Addressing Factors That Contribute to Fatigue and Sleepiness

Fatigue-inducing factors such as long work hours, irregular work shifts, a high number of miles driven, and violations of HOS regulations can have a critical bearing on truckers’ sleep patterns. Despite changes in HOS legislation to promote sleep and rest periods, the implications of these findings suggest the need for the continuous monitoring of CMV driver regulations and operational conditions [27].

It is widely acknowledged in the literature that the trucking industry needs outreach and intervention initiatives aimed at impacting driver behaviors to improve long-term health outcomes [28,33,99,100,101,102,103]. Researchers have offered recommendations and best practices as to what a CMV driver occupational health program should entail [15,28,37,103,104,105,106]. Characteristics of an effective program include high levels of instrumental support, engendering trust, motivating language, and competent communication [101,104,106,107]. An important guiding principle for designing such programs is the need to provide holistic and comprehensive interventions that follow established best practices [28]. The strengths of fleet wellness programs include providing no-cost services to drivers, providing clinical and physical testing and individualized coaching, offering incentive-based rewards and driver support groups, prioritizing the privacy of health information, and establishing a positive rapport with drivers [24,37,102,103,105,106]. There is an ongoing need for further research regarding CMV driver health and safety [31,108] and it has been suggested that more research be directed toward changes in the work structure and framework itself [109].

## 9. Tools and Technologies to Identify and Mitigate Fatigued Driving in CMV Drivers

Fatigue detection technologies can be an important life-saving tool for fatigue-related crash avoidance and/or mitigation. Such technologies are becoming increasingly popular among fleet operations to assist in the detection of early-onset fatigue and interfacing with the driver to prevent crashes. Commercially available fatigue detection technologies use sensors to monitor and record a variety of measures, including physiological data, driver positioning and movement behaviors, and driver performance metrics (e.g., steering behavior, lane position, steering wheel pressure, etc.) [98]. As technologies advance, hybrid systems (i.e., combining multiple measures) result in a more refined and robust fatigue monitoring and detection system [98]. Fatigue detection technologies provide supplementary solutions which, combined with education, safety culture, and safety management techniques, address the increasing problem of CMV driver fatigue. Research supports implementing fatigue detection technologies in fleet and industry operations to predict, detect, and alert to operator fatigue [98]. Mabry et al. [98] concluded, from empirical studies using sound experimental and statistical techniques, that the most promising and effective technologies for detecting fatigue relied on tracking physiological signals and driver behaviors indicative of fatigue (i.e., eyelid movement and closure, blinking patterns, head positions).

Organizations may employ a variety of advanced system technologies aimed at mitigating or eliminating various driver-error elements, including fatigue and inattention. Understanding the implications of driver monitoring systems (DMS), advanced driver assistance systems (ADAS), and automated driving system (ADS) technologies on fatigue detection and fatigue-related crashes is imperative, as technologies and system capabilities are advancing rapidly.

Early iterations of DMS in consumer vehicles utilized a variety of time-based and vehicle kinematic-based data to indicate fatigue in drivers (e.g., time with vehicle on, changes in steering wheel input), but systems that detect fatigue in situ and provide feedback to the driver have not become widely incorporated into fleet vehicles until recently. Advancements in camera-based methods for processing driver fatigue detection and drowsiness have allowed for a more procedural means to detect fatigue as well as provide feedback to a driver before a critical incident occurs. Relevant indicators may include changes in facial expressions, driving postures, eye-blink behavior, and saccadic movements [110]. Additionally, many available aftermarket DMS offer the detection of driver inattention [98,111]. Critical incidents related to fatigue may be mitigated by the presence of ADAS, as well as the current deployment stages of ADS technologies. The basic premise of ADAS/ADSs is to control some element of the dynamic driving task, typically steering or braking. ADAS may control the vehicle under normal operating conditions, as seen with adaptive cruise control and lane-keep assist technologies.

Future ADS deployment strategies seek to remove the driver from the vehicle entirely, and various iterations of marketplace deployment have been theorized and are being tested for feasibility. While true driverless systems remove any element of driver fatigue from the equation, other roles may be developed that would supplant the need for a driver, replacing them with an out-of-vehicle support monitor whose role would be to facilitate the safe operation of multiple heavy vehicles. Similar concerns exist for these monitors, as they are further removed from the vehicle environment and driving stimuli and may exhibit additional burdens of fatigue. Some developers even seek to simulate a driving environment by having support monitors operating in virtual reality, which may have its own implications on longitudinal operation. Evolving and adapting HOS regulations with the deployment of ADS-equipped heavy vehicles must also be considered.

## 10. Safety Programs and Culture to Mitigate Fatigue in CMV Operations

Unlike the traditional approach in mitigating fatigue (e.g., HOS or work scheduling), a fatigue risk management system (FRMS) has been discussed as having the potential to improve safety in the transport sector. Borrowing from the aviation industry [112], an FRMS is basically the comprehensive and on-going monitoring of data (fatigue causal variables) and the evaluation of relevant consequences within a given context [113]. Data can later be used as a basis for appropriate fatigue–risk mitigation strategies and continuous improvement, such as the implementation of a program such as the North American Fatigue Management Program. While this approach has been promising, successful implementations in the workplace may require a lot of effort and can be considerably challenging [114], particularly to fleets with limited resources. Various measures for the monitoring and control of fatigue risks have been discussed in the literature [115].

Organizations also need to develop a culture that recognizes driver and organizational factors that contribute to the development of fatigue, realign programs and policies to reduce or eliminate factors known to contribute to driver fatigue, promote self-reporting of fatigue and dispatching best practices associated with reducing fatigue, educate drivers and their support systems on healthy sleep habits, train drivers to recognize early signs of fatigue, and implement programs to screen and treat sleep disorders [116,117]. Although developing a strong, positive safety culture takes time, it is possible with dedication and commitment from organizational management and executives [118].

## 11. Conclusions

The adverse impacts of fatigue not only affect the well-being of CMV operators but degrade their safe driving ability and impact the safety of others sharing the roadway. It is clear that lifestyle and behavioral factors common among truck drivers contribute significantly to the fatigue that plagues the trucking industry. Long working hours, irregular sleep and work schedules, and lack of accessibility to healthy food and opportunities for exercise create ideal conditions for obesity and related comorbidities that impact fatigue. Industry practices to detect and mitigate fatigue and subsequent crash risk through fatigue assessments, physiological measurements, and driver monitoring are important; however, further research is needed to address the root causes of fatigue, such as systemic industry practices or work culture that unintentionally contribute to fatigue and sleepiness. Fatigue detection technologies provide supplementary solutions which, when combined with education, safety culture, and safety management techniques, are important steps to address CMV driver fatigue. Overall, outreach and intervention, technology-based or otherwise, is needed to disrupt the trend of unhealthy work culture in the trucking industry, which often leads to increased crash risk. In summation, preventing fatigue is imperative for safe commercial driving.

## Figures and Tables

**Table 1 ijerph-19-14780-t001:** Keywords and search terms used in narrative review.

Fatigue	Truck Driver	Health(y)	Crash	Safety Culture
Sleep	Commercial driver	Wellness	Accident	Health program
Drowsiness	Commercial motor vehicle driver	Obesity	Motor vehicle crash	Safety program
Sleepiness	Trucking	Obstructive Sleep apnea	Driving	Advanced driver assistance
Shiftwork	Occupational drivers	Sleep apnea	Crash risk	Risk management
Impairment		Stress	Safety	
Work environment		Lifestyle		
Work hours		Behavioral factors		
		Dietary impact		
		Autonomic nervous system		
		Vital signs		

## Data Availability

Not applicable.

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
