# Peer review of "Unravelling the Complexity of Irregular Shiftwork, Fatigue and Sleep Health for Commercial Drivers and the Associated Implications for Roadway Safety"

_ijerph, 2022, doi:10.3390/ijerph192214780_

Round 1
Reviewer 1 Report
Dear authors, thank you for the opportunity to read your review article.
Unfortunately, your article cannot be recommended for publication for the following reasons:
The structure of the article does not have all the necessary sections (introduction, materials and methods, results, discussion of the results and conclusions). The design of literature sources is done with errors. There are 22 sources in total for the last 5 years, which is very small, given that the article is a review article.
The introduction indicates the relevance of the study. At the same time, the authors do not clearly spell out the goal, hypothesis and theoretical model that the authors describe. From this it is not clear on what basis articles were selected for analysis and subsections of the theoretical review were made. The structure of the subsections is unclear and not justified. There is no discussion of the results, which would help clarify why the analysis was done, how the authors plan to use these results in further empirical research.
A number of review articles are built on the principle of analyzing publications for a certain period of time on the stated topic, and then their content is analyzed, the similarity of the problems under study and conclusions, etc. In this case, there is no such description either, and therefore the purpose of the theoretical study is unclear and incomprehensible.
I hope that the recommendations made will allow you to revise your manuscript and submit it again.
Best regards, the reviewer.
Reviewer 2 Report
Dear authors,
I carefully read your manuscript, which aims to review literature about fatigue and sleepness in workers in the transport sector.
I have some doubts, suggestions, and curiosities, which, I hope, will improve the quality of your article.
My major concerns are
1. The aim and scope of your manuscript are unclear and, in my opinion, do not wholly reflect the manuscript's content. In re-defining the scope of your review and the answer you intend to respond with this article, I suggest using the PICO.
2. The typology of your review is not specified: is it a narrative, systematic, meta-analysis or what else?
3. Materials and methods are entirely missing. You should report the databases you searched, the keywords you used, the time frame, a flowchart of the found and included/excluded articles and other essential information needed in a review.
4. In line with the previous comment, results and discussion sections are missing.
5. In order to better understand the contents of the manuscript, it would be helpful for the reader to introduce and explain the chapters' division; otherwise, the way the article is now is a little confusing and difficult to follow.
6. Lines 68-74: in the introduction section, you never mentioned the lifestyle and health of this population; however, these aspects are mentioned in the second paragraph. Maybe it is better to revise the introduction and focus more on the aspects explained in the manuscript's chapters.
7. It would be helpful for the readers to add some conclusion sentences at the end of each chapter to resume and better understand each topic's state of the art.
8. In my opinion, the conclusion section should be revised: what does it add to the previous literature?
Please, see later for minor comments:
1. Line 36: the citation is missing.
2. Did you register your review on PROSPERO? In the negative case, I suggest doing it.
3. Lines 30 and 78: I suggest adding the following references:
doi: 10.3389/fphys.2020.00693; doi: 10.1111/jonm.12756; and doi: 10.1016/j. jsmc.2009.03.001
4. It would also be interesting to deeper or to dedicate some lines more to exercise and physical activity. Physical activity practice could have some influences on fatigue.
5. The way the citations are reported in the text are several and different from each other: sometimes the research group is reported with "et al.," and sometimes with "and colleagues"; sometimes the year is within brackets, while sometimes it is missing. I suggest checking every citation and standardizing them
6. Line 92: the citation is missing here; I suggest the following articles: doi: 10.3389/fphys.2021.628231, doi: 10.3390/ijerph18168378; doi: 10.1007/0-387-23692-9_17; doi: 10.1093/ sleep/30.11.1460 and doi: 10.1093/occmed/kqg046
7. PAP abbreviation is not introduced.
8. Chapter 6: is actigraphy a valid method? Are there any articles involving its use?
9. I do not understand the meaning and the necessity of chapter 7.
10. Chapters 9 and 11 seem to speak about the same topic. Could you merge them or, alternatively, better differentiate them?
Round 2
Reviewer 1 Report
Dear authors, I have carefully studied all your comments. I can agree that the format of your article is special. If the editors of the Journal and the special issue allow it, there is no problem. At the same time, a number of remarks have not been corrected:
adding sources of literature for the last 5 years and adjusting the design of the references in accordance with the requirements of the Journal. I think that these two comments could be taken into account.
Best regards, Reviewer
Author Response
Thank you for your comments and suggestions, which we have addressed in the revised manuscript:
- The authors added 25 recent (within the last 5 years) references to support the narrative review
- All references and in-text citations were formatted to meet the journal requirements (i.e. numbered in order of appearance in the text and listed individually at the end of the manuscript. In the text, reference numbers should be placed in square brackets [ ], and placed before the punctuation).
Reviewer 2 Report
Even though the narrative nature of the review, I suggest adding the databases used for the articles research, keywords used for the article research, and the years within the articles have been selected.
Author Response
Thank you for the thoughtful feedback and revision suggestions. The authors have made the following revisions:
- Lines 42 and 43 added to note databases used for the literature search
- Table 1 added (pg. 2/14) to list keywords used for the literature search
- Lines 43 and 44 added to include years of publications included in narrative review.